# Mining Math Conjectures from LLMs: A Pruning Approach

**Jake Chuharski**[1*]    **Elias Rojas Collins**[1*]    **Mark Meringolo**[2]

[1]Massachusetts Institute of Technology    [2]Unaffiliated

{chuharsk,erojasc}@mit.edu

markpmeringolo@gmail.com

## Abstract

We present a novel approach to generating mathematical conjectures using Large Language Models (LLMs). Focusing on the solubilizer, a relatively recent construct in group theory, we demonstrate how LLMs such as ChatGPT, Gemini, and Claude[1] can be leveraged to generate conjectures. These conjectures are pruned by allowing the LLMs to generate counterexamples. Our results indicate that LLMs are capable of producing *original* conjectures that, while not groundbreaking, are either plausible or falsifiable via counterexamples, though they exhibit limitations in code execution.

## 1   Introduction

Artificial intelligence, specifically deep learning, has created much discussion around the possibility to augment human creativity with computational capability. Among the leading technologies pushing this discussion are large language models (LLM's) such as OpenAI's ChatGPT, Anthropic's Claude, and Google's Gemini (1; 2; 3). While LLMs have been widely recognized for their competence in text generation, their interactions within abstract academic fields such as mathematics, specifically with conjecture creation, remain under-explored. Initial work has evaluated LLMs' ability to pass exams like the SAT and MBA qualifying exams (4; 1). More recently, efforts have focused on bench-marking their capacity to generate mathematical proofs (5). However, there has been little work on bench-marking the ability of language models to act as a creative agent towards coming up with new conjectures.

In this study, we use the Claude Sonnet, Gemini 1.5, and GPT-4 APIs to both generate conjectures and write GAP computer algebra code to check them for plausibility. GAP (Groups, Algorithms, and Programming) is a computer algebra system designed for computational group theory and related areas in abstract algebra. However, GAP is not a proof assistant so does not give the user proofs for theorems, but it can be used to check conjectures for immediate counterexamples. We work specifically on the solubilizer subset which is a relatively new/unexplored construction in group theory that contains much potential for novel conjectures (see Appendix A.1). GAP computer algebra can check the conjecture for chosen groups and allows for language models to "guess and check". The system provides a method to mine for conjectures using language models and a pruning step to remove conjectures that are false for obvious (or sometimes non-obvious) reasons. This approach offers a systematic method for generating and validating conjectures, combining model output with automated computational verification without requiring a strong formal theorem prover.

---

[*]These authors contributed equally to this work

[1]We additionally add information from a small sample on OpenAI o1 preview in Appendix A.5

38th Conference on Neural Information Processing Systems (NeurIPS 2024).

## 2   Related Work

Recent studies have explored LLMs' role in conjecture generation. Johansson and Smallbone observe that many of the symbolic structures generated by LLMs may already exist in training data, raising concerns about genuine originality (6). They note that GPT-4 appears to have been trained on proof libraries like QuickSpec, Hipster, and Isabelle/HOL, providing a potential caveat for verifying the originality of any generated conjectures.

We mitigate this challenge by deliberately focusing on a mathematical area with limited prior exposure: the solubilizer (see Appendix A.1). By iteratively updating prompts, we also attempt to steer the models away from generating *as many* redundant conjectures which they also found to be a problem because "GPT-4 usually produces the same kind of 'generic' lemmas every time" (6).

Other studies, such as Davies et al. (7), use machine learning to assist mathematicians in proof creation rather than conjecture generation. In Wu et al. (8), LLMs are shown to autoformalize natural language math into formal theorem provers like Isabelle, translating competition problems into formal proofs with impressive accuracy. In Si et al. (9), LLMs are evaluated on the ability to be creative agents in coming up with research ideas, however math was touched minimally. They additionally corroborate the claim that "LLMs lack diversity in idea generation"(9). These approaches focus on proof generation, formalization, or assistance, whereas our work emphasizes the initial creative step of formulating new conjectures, and then provides an immediate 'guess-and-check' step to verify plausibility.

## 3   Methodology

The method that we propose to "mine" for math theorems, shown in Table 10, is as follows:

1. We begin with a prompt that includes literature on the solubilizer from (10; 11; 12; 13; 14; 15; 16). The model is prompted to generate theorems related to the literature provided and write GAP code to test conjectures on groups. Full prompting is provided in Appendix A.2.
2. The LLM then generates GAP code and the GAP code is run.

   ✓ If the code compiles and runs and the outcome is recorded.

   ✗ If the code does not compile the LLM is prompted again to fix the code, provided with the output of the failing program. It is given the chance to do two revisions (in practice allowing for further revision almost never results in working code).

3. If the result is that the conjecture is false, the theorem and it's result is added to the prompt, and the process is repeated with the false conjecture added to the set of ideas that are known to fail.

This process was run with three models: ChatGPT 4 (`gpt-4o-2024-05-13`), Claude Sonnet (`claude-3-5-sonnet-20240620`), and Gemini 1.5 (`gemini-1.5-flash`). LLM's have a "Temperature" parameter which varies the level of randomness in the outputs to a given prompt. This is sometimes taken as a proxy for "Creativity", although this description is disputed (17). The temperature for the Claude model was set to 1 for conjecture generation and .1 for code generation. The GPT-4 conjecture was set to 1.08 for conjecture generation and was left at default for code generation. The Gemini 1.5 conjecture generation was set to 1.5 (`top_k: 5`, `top_p:.99`) and default for code generation. The values were generated by trial and error where the authors observed qualitatively the most consistent conjecture variation without extreme hallucinations[2].

### 3.1   Area of Focus

The mathematical area of focus is called the *solubilizer* and is defined as follows:

**Definition 3.1.** *Let $G$ be a finite group. For any element $x \in G$, the solubilizer of $x$ in $G$ is defined as:*

$$\mathrm{Sol}_G(x) := \{y \in G \mid \langle x, y \rangle \text{ is soluble}\}.$$

More introductory and historical information on the solubilizer can be found in the Appendix A.1.

---

[2]For example, if the temperature is set too high in GPT-4, the model will return output in multiple languages

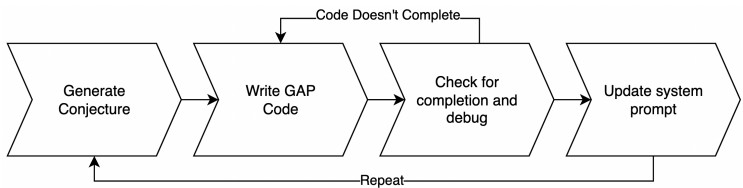

Figure 1: Method

# 4 Results

## 4.1 Performance Overview

The experiment provided three types of outcomes (Summarized in Table 3):

- Successful generation of counterexample-finding code in 25.95% of cases (109 out of 420 unique outputs).
- Generation of conjectures without counterexamples in 9.52% of cases (40 out of 420 unique outputs).
- Generation of non-executable code in 64.52% of cases (271 out of 420 unique outputs).

Table 1: Classification of Outputs

| Category | ChatGPT | Claude | Gemini | Total |
|----------|---------|--------|--------|-------|
| Unique Conjectures | 94 | 89 | 237 | 420 |
| Total Output | 249 | 258 | 250 | 757 |
| No Counter-Examples | 25 | 4 | 11 | 40 |
| Couldn't Execute Code | 33 | 44 | 194 | 271 |
| Conjecture Failed | 36 | 41 | 32 | 109 |

## 4.2 Examples

The following is an example with no counter-examples from Claude:

**Conjecture 4.1.** *Let $G$ be a non-solvable group. For any element $x$ in $G$, if $\mathrm{Sol}_G(x)$ is a subgroup, then the Frattini subgroup of $\mathrm{Sol}_G(x)$ is contained in the Frattini subgroup of $G$.*

This result is simple enough that the model can be prompted to prove the conjecture with slight modification. See appendix A.4.1. The following is a conjecture that failed from Gemini (see A.4.2):

**Conjecture 4.2.** *Let $G$ be a non-solvable group, and let $x \in G$. If $|Sol_G(x)|$ is divisible by exactly two primes, then one of them is 2.*

**Output 4.3.** *Conjecture failed for group:* $\mathrm{PSL}(3, 2)$

Where $\mathrm{PSL}(3, 2)$ is the projective special linear group of 3x3 matrices over the finite field $\mathbb{F}_2$. Lastly, we have an example where code could not be executed from GPT-4:

**Conjecture 4.4.** *Let $G$ be a finite non-solvable group and $x \in G$. Then for every abelian subgroup $A$ of $G$, we have $\mathrm{Sol}_G(x) \cap A \neq \{1\}$.*

## 4.3 Similarity Analysis

To quantitatively measure diversity, we calculated the cosine self-similarity, and similarity between conjecture sets/literature. The similarity results are summarized in Table 2. Where we see a maintained level of similarity throughout the experiments and between models. See Appendix A.7 for heatmaps.

Scaling the system further by increasing the number of trials or generating more variations in the prompt failed to yield significantly more diverse conjectures. However the quality did not decrease either. This finding suggests that a different approach, such as multi-modal model interaction or combining LLMs with automated theorem provers, could help diversity.

Table 2: Output Cosine Similarity Distribution

| Metric | Max | Min | Mean | Median |
|--------|-----|-----|------|--------|
| **ChatGPT vs. ChatGPT** | 0.9284 | 0.0486 | 0.2911 | 0.2658 |
| **Claude vs. Claude** | 0.9950 | 0.0921 | 0.4238 | 0.3683 |
| **Gemini vs. Gemini** | 0.8742 | 0.0132 | 0.2451 | 0.2264 |
| **ChatGPT vs. Claude** | 0.8695 | 0.0356 | 0.2699 | 0.2552 |
| **ChatGPT vs. Gemini** | 0.8892 | 0.02638 | 0.2442 | 0.2292 |
| **Claude vs. Gemini** | 0.7662 | 0.0149 | 0.2220 | 0.2102 |
| **Claude vs. Literature** | 0.7101 | 0 | 0.1501 | 0.1332 |
| **Gemini vs. Literature** | 0.8274 | 0 | 0.1812 | 0.1510 |
| **GPT-4 vs. Literature** | 0.9388 | 0 | 0.2139 | 0.1978 |

# 5 Discussion

## 5.1 Observations

Among the 757 outputs generated by the LLMs, 420 unique conjectures were identified. The high number of duplicates shows a considerable redundancy in the results, as approximately 55.48% of the conjectures were deemed to be unique. While not unexpected, the duplicates suggest that LLMs likely rely on similar patterns when prompted similarly across trials as described in (6). However, this did not significantly hinder overall performance other than increase the number of total iterations needed to yield a desirable number of conjectures.

Not all of the conjectures generated by the models were entirely original and was verified by one of the authors of the seven original solubilizer papers for all 420 unique conjectures. For example,

**Theorem 5.1.** *Let G be an insoluble group and x an element of G. Then the cardinality of cannot be equal to $p^2$ for any prime p.*

shows up in (12) and GPT-4 conjectured:

**Conjecture 5.2.** *Let G be an insoluble group and $x \in G$. Then the cardinality of $\mathrm{Sol}_G(x)$ cannot be equal to $p^2$ for any prime p.*

That being said, this result was contained in the system prompt and can be ignored. In all other cases the models output conjectures that were distinct from anything found in literature or their system prompt.

In 109 cases (25.95%), the generated code successfully identified counterexamples, which is critical for falsifying conjectures. Secondly, of the 420 unique outputs, only 40 (9.52%) produced conjectures with no counterexamples. ChatGPT significantly outperformed both Claude and Gemini in this area, generating 26.60% valid conjectures compared to Gemini's 4.64% and Claude's 4.49%. This shows that ChatGPT was more effective at producing conjectures that are plausible at first glance. However, a large portion of the GPT-4 conjectures were looking at the size of the solubilizer rather than about interactions with other groups, group structure, or subgroup properties. Therefore, one could argue that they were easier to write code for, or at least more likely to succeed based on similarity. Further still, results classified as having "no counterexamples" by GPT-4 seemed to be qualitatively more obvious than those by Claude or Gemini (see Conjecture A.5 vs. Conjecture A.4 vs. Conjecture A.9). Lastly, the fact that the models are able to generate novel, original conjectures at all provides promise for these models to be used as useful tools when developing the theory of a new construction.

## 5.2 Limitations

A limitation observed in both models was the generation of non-executable code, which occurred in 271 instances (64.52% of unique outputs). Gemini and Claude struggled more with code execution, having 81.86% and 49.44% instances of non-executable code respectively, compared to ChatGPT's 37.76%. This potentially points to differences in how the models handle code syntax in GAP, however the models were prompted to have a near identical code format (see Appendix A.2). This corroborates the idea that some conjectures set by Claude/Gemini are more difficult to write code for, and therefore more likely to fail in this system. Interestingly, there are some examples that Claude/Gemini gave

that would be much harder to check in GAP with a constrained time limit where Claude/Gemini could not write executable code (see Appendix A.4.6). Lastly, we note that the models had different approaches to generating code to test conjectures, with Claude and Gemini being more similar. We found that ChatGPT liked to preemptively restrict the groups it would consider. For example, ChatGPT conjectured that the solubilizer couldn't be bigger than or that it couldn't be exactly equal to any of the following numbers (for all non-solvable groups) in seperate conjectures: [2, 3, 6, 8, 9, 10, 11, 12, 14, 15, 16, 18, 20, 24, 25, 27, 32, 49, 50, 126].

While this study focuses on group theory and solubilizers, a relatively unexplored area, the approach could be generalized to other domains. We acknowledge limitations of using GAP, an algebra software. However, future work could easily extend this methodology to fields like number theory, geometry, representation theory, or combinatorics by integrating tools like SageMath, MAGMA, or other computational solvers.

### 5.3 Future Work

We propose the investigation of conjecture generation in fields where existing conjectures are sparse or absent. For example, LLMs could be applied to generate conjectures in newer or less explored areas such as tropical geometry or higher homotopy theory, where automated tools exist but have yet to be fully integrated with LLMs. Furthermore, the study above was limited to using a single LLM. If one model is better at writing code and the other is better at conjectures, using a combination structure could yield better results. We remark that a quantitative metric for 'interestingness' of a math conjecture or problem seems to be elusive, nontrivial, yet useful (see Appendix A.6).

## 6    Conclusion

The study opens up several promising avenues for the use of LLMs in research. Our work, while small, shows the potentially impactful way that LLM's augmented with other computational capacity can solve more complex problem. For example, further work integrating conjecture generation with proof validation systems could streamline the process of discovery.

That being said, LLM-based conjecture generation is still very limited to existing knowledge. Rather than producing fundamentally new ideas, LLMs are likely to lean on known results, limiting their ability to drive groundbreaking discoveries (18). Indeed, when thinking of language models as statistical traversers of some sort of higher dimensional surface built from training data, it is easy to imagine that the models are not able to stray too far from what they are fed to generate the surface. Specifically, conjectures and theorems involving well-understood subgroups on which the solubilizer is inspired (think centralizer and normalizer) can serve as an incredibly large well from which an LLM can sample a new direction about the solubilizer. This may all be permissible to a practitioner if one is only interested in clearing out the brush around a new construct such as the solubilizer; but as of writing, it should not be expected that these models will conjecture something profound.

We demonstrate that combining LLMs with computational resources like GAP can successfully generate and test original, albeit simple math conjectures. Indeed, performance suggests that LLMs like ChatGPT, Claude, and Gemini have potential, but only on conjectures that are similar to existing ideas or are otherwise simple. Furthermore, the models face significant challenges in generating executable code and avoiding duplicate conjectures. Indeed, ChatGPT-4 demonstrated stronger performance in generating conjectures that could not be immediately falsified, Claude was slightly more effective at identifying counterexamples, and Gemini had the least redundancy likely due to the longer context window. The high percentage of non-executable code reinforces the need for robust error-checking and handling within the models. GAP is limited in the variety of error codes that are produced when code fails, so other more verbose computational algebra solvers could help with error correction. Lastly, further analysis of failed code generation to find patterns of failure could lead to better prompting for avoiding common bugs. Further work would likely include adding a formal automated theorem prover or another form of neuro-symbolic proof engine, giving an end-to-end system that can generate new conjectures and prove them in a single pass(19; 20). The authors are also interested to see other new approaches for accurate conjecture generation in various abstract fields, or more generally, improvements to conjecture generation by non-LLM based models.

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

# A  Appendix / Supplemental Material

## A.1  Solubilizer

### A.1.1  Background

The *solubilizer* is a construct that can be attributed to Doron Hai-Reuven in (14) from 2013. Much of the work in the area can be atributed to Banafsheh Akbari et al. in (10; 11; 12; 13). The construct itself should be intelligible for anyone who has taken an introductory university algebra course, although can be potentially unpredictable to work with. It is therefore also simple for the models to understand, but non-trivial to produce conjectures. All of the models claim to have not heard of the construction before, meaning that the training data either excluded this information or did not have enough for it to be in the knowledge base. In either case, the models seem to be able to understand the definition once it is given, and provide often accurate assumptions about how the set behaves. For example, when testing these ideas on GPT-4, the model responded with the following:

**Input:**    Give me the definition of the solubilizer in group theory

**Output:**    *"In group theory, there isn't a concept specifically called 'solubilizer.' However, I suspect you might be referring to the 'solubility' of a group or the concept of a 'solvable group.' A group $G$ is said to be solvable if there exists a sequence of subgroups*

$$\{e\} = G_0 \triangleleft G_1 \triangleleft G_2 \triangleleft \cdots \triangleleft G_n = G$$

*such that each $G_i$ is a normal subgroup of $G_{i+1}$ and the quotient group $G_{i+1}/G_i$ is abelian. Solvable groups generalize the notion of abelian groups. Solvability is an important property in group theory and plays a key role in Galois theory, which connects group theory with field theory. If you had a different concept in mind or need more information about solvable groups or another topic, please clarify, and I'll be happy to assist further!"*

We also further note that the model presents an almost identical response to "solvabilizer" instead of "solubilizer" which is another name that the set can be found under in the literature. We then present the definition to the model so that it can begin to formulate an understanding of the area.

**Input:**    I will define the solubilizer as follows: Let G be a finite group. For an element x in G, we define the solubilizer of x in G as the set: $SolG(x) := \{y \in G | \langle x, y \rangle$ is soluble$\}$

**Output:**    *"Ah, I see. Given your definition, the solubilizer $\mathrm{Sol}_G(x)$ of an element $x$ in $G$ is the set of all elements $y$ in $G$ such that the subgroup generated by $x$ and $y$, denoted $\langle x, y \rangle$, is a solvable group. To further elaborate on this:*

1. *$\langle x, y \rangle$ denotes the subgroup of $G$ generated by the elements $x$ and $y$. This is the smallest subgroup of $G$ that contains both $x$ and $y$.*
2. *The condition that $\langle x, y \rangle$ is solvable means that there exists a sequence of subgroups, starting with the trivial group and ending with $\langle x, y \rangle$, such that each subgroup in the sequence is a normal subgroup of the next, and the quotient of successive subgroups is abelian.*
3. *The solubilizer $\mathrm{Sol}_G(x)$ is then the set of all elements $y$ in $G$ for which this condition holds.*

*This is an interesting definition, and it essentially captures the elements in $G$ that, when paired with $x$, produce a solvable subgroup. The concept seems to be related to understanding the structure and solvability properties of subgroups within $G$. Do you have specific questions about this definition or its properties?"*

This output shows three correct assertions about the subset, so we test to see if the model can expand further on it's understanding without more input. The model is then asked to provide some conjectures based on the newly learned definition. The idea is to gauge the model's understanding of a single definition and to see what material the model relates to the new definition to conjecture something that it considers to be new material.

**Input:** Based on this definition, can you conjecture any theorems that might apply to this set?
**Output:** *"Certainly! Given the definition of the solubilizer $\mathrm{Sol}_G(x)$ in a finite group $G$, we can propose some conjectures or theorems. Here are a few ideas:*

1. **Non-Empty Solubilizer**: For any element $x$ in $G$, $x$ belongs to $\mathrm{Sol}_G(x)$ because the subgroup $\langle x \rangle$ (generated solely by $x$) is cyclic and hence solvable.

   **Theorem**: For any $x \in G$, $x \in \mathrm{Sol}_G(x)$.

2. **Subgroup Containment**: If $y$ belongs to $\mathrm{Sol}_G(x)$, then $y^{-1}$ (the inverse of $y$) also belongs to $\mathrm{Sol}_G(x)$. This is because if $\langle x, y \rangle$ is solvable, then $\langle x, y^{-1} \rangle$ is also solvable.

   **Theorem**: For any $y \in \mathrm{Sol}_G(x)$, $y^{-1} \in \mathrm{Sol}_G(x)$.

3. **Closure under Solvability**: If $y_1, y_2 \in \mathrm{Sol}_G(x)$ and their product $y_1 y_2$ belongs to $\langle x, y_1 \rangle$ or $\langle x, y_2 \rangle$, then $y_1 y_2$ should also belong to $\mathrm{Sol}_G(x)$.

   **Theorem (Conditional)**: For $y_1, y_2 \in \mathrm{Sol}_G(x)$, if $y_1 y_2 \in \langle x, y_1 \rangle$ or $y_1 y_2 \in \langle x, y_2 \rangle$, then $y_1 y_2 \in \mathrm{Sol}_G(x)$.

4. **Entire Group Solvability**: If $G$ itself is solvable, then for any $x \in G$, $\mathrm{Sol}_G(x) = G$.

   **Theorem**: If $G$ is solvable, then for all $x \in G$, $\mathrm{Sol}_G(x) = G$.

5. **Size of the Solubilizer**: The size of $\mathrm{Sol}_G(x)$ provides information about the complexity of the group's structure in relation to solvability.

   **Conjecture**: For certain classes of groups (e.g., simple groups), the size of $\mathrm{Sol}_G(x)$ relative to the order of $G$ may have particular bounds or properties.

All five of these ideas are true. The first four are easily proved to be correct mathematical statements and the fifth is just a general statement that presents the (correct) expectation that the solubilizer may be informative to studying the structure of the group.

### A.1.2  Math

Let $G$ be a finite group, and for any element $x \in G$, the solubilizer of $x$ in $G$ is defined as:

$$\mathrm{Sol}_G(x) := \{y \in G \mid \langle x, y \rangle \text{ is soluble}\}.$$

In general, $\mathrm{Sol}_G(x)$ is not necessarily a subgroup of $G$. However, there are specific conditions under which this set does form a subgroup. It has been proven in (14) that $\mathrm{Sol}_G(x)$ is a subgroup of $G$ for any element $x \in G$ if and only if $G$ is a soluble group.

A well-known result, attributed to Thompson (21), states that a finite group $G$ is soluble if and only if, for every $x, y \in G$, the subgroup $\langle x, y \rangle$ is soluble. Thus, a finite group $G$ is soluble if and only if for any element $x \in G$ $\mathrm{Sol}_G(x) = G$.

An important related concept is the soluble radical $R(G)$, the largest soluble normal subgroup of $G$. Guralnick et al. (22) demonstrated that for an element $x \in G$, $x \in R(G)$ if and only if $\langle x, y \rangle$ is soluble for all $y \in G$. Consequently, $x \in R(G)$ if and only if $\mathrm{Sol}_G(x) = G$.

A common question is how the structure of a single solubilizer influences the structure of the entire group. For instance, it was shown in (14) that if $G$ contains an element $x$ such that all elements of $\mathrm{Sol}_G(x)$ commute pairwise, then $G$ must be abelian. Another example, (11) generalizes this by proving that if there exists $x \in G$ such that for every $u_1, u_2, u_3 \in \mathrm{Sol}_G(x)$, the commutator $[u_1, u_2, u_3] = 1$, then $\gamma_3(G) = 1$, implying that the group is nilpotent. Here, $\gamma_3(G)$ is the third term in the lower central series of $G$. These are ideas that are more often explored by Claude and Gemini, although are harder to verify.

The arithmetic properties of solubilizers also play a crucial role in determining group structure. For example, if $G$ contains an element $x$ whose solubilizer has order $p$ or $p^2$ (where $p$ is a prime), then $G$ is a $p$-group, as discussed in (12). These ideas seem to be of heavy focus for ChatGPT.

Thompson's theorem (21) demonstrated that a finite group $G$ is soluble if and only if every two-generated subgroup of $G$ is soluble. This motivates the definition of the solubilizer and highlights its significance. Moreover, the solubilizer operates similarly to the centralizer in a group, with the soluble radical $R(G)$ functioning in a way comparable to the center of a group. Analogous to the centralizer $C_G(x) = \{y \in G \mid \langle x, y \rangle \text{ is abelian}\}$, the solubilizer describes solvability rather than commutativity. However, unlike the centralizer, the solubilizer is not always a subgroup which was initially tough for the models to rememeber, leading to this fact requiring repetition in the prompting.

## A.2 Model Usage and prompting

Both the prompt for generating conjectures and the prompt for generating code were prefaced with:

```
1    """You are a mathematician and efficient computer scientist. You
     are interested in abstract algebra, but are generally very
     knowledgeable and interested in the intersections between
     different areas of math.
2
3    You have began working on the 'solubilizer' subset of a non-
     solvable group. You are very good at writing GAP code. The user
     will ask for either a conjecture, or GAP code to check a
     conjecture.
4
5    When a user asks for GAP computer algebra code you will provide
     nothing in your response except code to complete their task. When
     writing code make sure that the answer to your question is PRINTED
      to the terminal.
6
7    A maximum of two things should be printed by your code. If the
     conjecture fails, the code should break and just print for which
     group the conjecture failed. If the code does not generate any
     counter-examples, the code should return "No Counter-examples!".
8
9    Additionally, make sure you only test conjectures on Non-Solvable
     groups! For example, when checking a conjecture, you might want to
      check it on all non-solvable groups of order less than one
     million by using: SimpleGroupsIterator(1, 10^6). This means ONLY
     give GAP computer algebra code unless the user asks for a
     conjecture. When the user asks for a conjecture you should return
     nothing but the conjecture."""
```

The code generation system prompt for both models included code snippets for how to accurately and consistently generate the solubilizer and had the format for how the conjectures should be checked and output.

```
1    "The general function for the solubilizer in GAP is here, you will
         have to write the rest of the checks yourself":
2        solubilizer := function(G, max, x)
3            rad := RadicalGroup(G);
4            if x in rad then
5                return G;
6            else
7                M := List(max);
8                M := Set(Filtered(M, m -> x in m));
9                maxes := [];
10               solx := [];
11               while Size(M) > 0 do
12                   m := M[1];
13                   if IsSolvable(m) then
14                       solx := Union(solx, List(m));
15                       maxes := Union(maxes, [m]);
16                       Remove(M, 1);
17                   else
18                       MM := MaximalSubgroups(m);
19                       MM := Set(Filtered(MM, mm -> x in mm));
20                       Append(M, MM);
21                       Remove(M, 1);
22                       M := Set(M);
23                   fi;
24               od;
25               return solx;
26           fi;
27       end;
28   "When you test a conjecture. Make sure to have the end of your code be
         in this format":
```

```
29 CheckConjecture := function ()
30     local G, gen, x, solGx;
31
32     for G in SimpleGroupsIterator (1, 1000000) do
33         [PUT THE CODE TO CHECK THE CONJECTURE HERE!]
34         Print("Conjecture failed for group: ", StructureDescription (G)
       , "\n");
35         return;
36     od;
37     Print("No Counter-examples!\n");
38 end;
39 CheckConjecture ();
```

Listing 1: Solubilizer Prompting

The system prompt for generating conjectures also included information from the literature that was left in LaTeX form, including the subset's definition as above. For brevity they will not all be listed here but were in the form of:

```
1 Let \(G\) be a group. Then \(o(x)\) divides \(|\operatorname{Sol}_G(x)
    |\) for all \(x \in G\).
2
3 Let \(G\) be a group. Then \(|C_G(x)|\) divides \(|\operatorname{Sol}
    _G(x)|\) for all \(x \in G\).
```

At the end of the included known true conjectures, the system was also able to continuously update a set of conjectures falsified by the model.

```
1     The following are conjectures that you know to be false:
2 Let \( G \) be a finite non-solvable group and \( x \in G \). Then \(
    |\operatorname{Sol}_G(x)| \) is always an even number.
3 Let G be a non-solvable group. For any element x in G, if Sol_G(x) is
    a subgroup of G, then Sol_G(x) is nilpotent.
4 Let G be a non-solvable group. For any element x in G, if Sol_G(x) is
    a subgroup of G, then the derived subgroup [Sol_G(x), Sol_G(x)] is
     contained in the Fitting subgroup of G.
5 Let G be a non-solvable group. For any element x in G, the index of
    Sol_G(x) in its normalizer N_G(Sol_G(x)) is always a prime power.
6 Let G be a non-solvable group. For any element x in G, if Sol_G(x) is
    a subgroup, then the derived length of Sol_G(x) is strictly less
    than the derived length of G.
```

### A.2.1 Compute

The computer used for the experiments has the following specifications:

- Model: Macbook Air
- Chip: Apple M1
- Memory: 8Gb
- OS: MacOS BigSur 11.6 (20G165)

The experiments took between 48-72 hours to run for each model. This was mainly due to checking all non-solvable (or in some cases just simple) groups of order up to 1,000,000.

### A.3 Additional Examples

In the following we include one example of each type from each model.

### A.3.1 Claude

Example with no counterexamples from Claude:

**Conjecture A.1.** *Let $G$ be a non-solvable group. For any two elements $x, y \in G$, if $\mathrm{Sol}_G(x) \cap Sol_G(y)$ is non-empty, then $\mathrm{Sol}_G(x) \cap Sol_G(y)$ contains a non-trivial normal subgroup of $G$.*

The following conjecture failed:

**Conjecture A.2.** *Let $G$ be a non-solvable group. For any element $x$ in $G$, if $\mathrm{Sol}_G(x)$ is a subgroup of $G$, then the derived subgroup $[\mathrm{Sol}_G(x), \mathrm{Sol}_G(x)]$ is contained in the Fitting subgroup of $G$.*

**Output A.3.** *Conjecture failed for group: $A_5$*

In a similar example, the model could not write code that executed properly:

**Conjecture A.4.** *Let $G$ be a non-solvable group. For any element $x$ in $G$, if $\mathrm{Sol}_G(x)$ is a proper subgroup of $G$, then the intersection of $\mathrm{Sol}_G(x)$ with its normalizer in $G$ is always properly contained in the normalizer of the Fitting subgroup of $G$.*

### A.3.2 GPT-4

The following had no counterexamples from ChatGPT 4:

**Conjecture A.5.** *For any finite non-solvable group $G$ and any element $x \in G$, the set $\mathrm{Sol}_G(x)$ is not a cyclic group.*

Similarly GPT-4 suggested that the following be true although it was immediately obvious to be false:

**Conjecture A.6.** *For any element $x \in G$ of a non-solvable finite group $G$, the set $\mathrm{Sol}_G(x)$ contains all elements of a certain conjugacy class in $G$*

**Output A.7.** *Conjecture failed for group: $A_5$*

Where $A_5$ is the alternating group on five elements (see A.4.4). Lastly, an example where GPT-4 could not execute code for the conjecture:

**Conjecture A.8.** *For any finite non-solvable group $G$, there exists an element $x \in G$ such that $\mathrm{Sol}_G(x)$ is a nilpotent subgroup of $G$.*

### A.3.3 Gemini

The following is a conjecture with no counterexample:

**Conjecture A.9.** *Let $G$ be a finite non-solvable group and suppose $x \in G$ is not an element of the soluble radical $R(G)$ of $G$. Assume that $\langle x, x^y \rangle$ is not solvable for any element $y \in G$. Then $\langle x, Sol_G(x) \rangle = Sol_G(x)$ for all $x \in G$.*

The following is a conjecture that is false:

**Conjecture A.10.** *Let $G$ be a finite non-solvable group. For any element $x$ of $G$, the probability that a randomly chosen element $y \in G$ is contained in $Sol_G(x)$ is less than or equal to the probability that $y$ is contained in the radical of $G$.*

**Output A.11.** *Conjecture failed for group: $A_5$*

Where, again, $A_5$ is the alternating group on $5$ elements (see A.4.5). The following is a conjecture where code could not be executed:

**Conjecture A.12.** *Let $G$ be a finite non-solvable group. Let $x, y$ be two non-commuting elements of $G$ such that the subgroups generated by $x$ and $y$, $\langle x, y \rangle$, is solvable. Then the probability of finding a third element, $w \in G$, that commutes with both $x$ and $y$, subject to the additional condition that at least one of the two groups $\langle x, w \rangle$ or $\langle y, w \rangle$ is solvable must be equal to at most the product of probabilities of a non-commutation of $x$ and $y$ and the existence of $w$ that commutes with $y$ and $x$.*

## A.4 Conjectures

### A.4.1 Proof of Conjecture 4.1

**Proof:** By assumption, $\mathrm{Sol}_G(x)$ is a subgroup of $G$. Let $\Phi(G)$ and $\Phi(\mathrm{Sol}_G(x))$ denote the Frattini subgroups of $G$ and $\mathrm{Sol}_G(x)$, respectively. We aim to prove that

$$\Phi(\mathrm{Sol}_G(x)) \subseteq \Phi(G).$$

The Frattini subgroup $\Phi(H)$ of a group $H$ is the intersection of all maximal subgroups of $H$. In particular, for any maximal subgroup $M$ of $\mathrm{Sol}_G(x)$, there exists a maximal subgroup $N$ of $G$ such that $M \subseteq N$. Since

$$\Phi(\mathrm{Sol}_G(x)) = \bigcap \{M \mid M \text{ is a maximal subgroup of } \mathrm{Sol}_G(x)\},$$

we have

$$\Phi(\mathrm{Sol}_G(x)) \subseteq \bigcap \{N \mid N \text{ is a maximal subgroup of } G\} = \Phi(G).$$

Thus, we conclude that $\Phi(\mathrm{Sol}_G(x)) \subseteq \Phi(G)$. □

### A.4.2 Failure of Conjecture 4.2

- Conjecture failed for group: $\mathrm{PSL}(3, 2)$
- Element: $(2, 8, 4, 3, 6, 7, 5)$
- Prime divisors of $|Sol_G(x)|$ : $[3, 7]$

### A.4.3 Failure of Conjecture A.2

- Conjecture failed for group: A5
- Element: (1,5,2,4,3)
- Derived subgroup: Group( [ (1,5,2,4,3) ] )

### A.4.4 Failure of Conjecture A.6

- Conjecture failed for group: $A_5$
- Conjugacy class: (3 4 5)
- Co-generator: (1 2 3)
- Generated group: $\langle (1\ 2\ 3), (3\ 4\ 5) \rangle = A_5$ is not solvable

### A.4.5 Failure of Conjecture A.10

- Conjecture failed for group: $A_5$
- Element: () (the identity element)
- Probability($\mathrm{Sol}_G(x)$) : 1
- Probability(Radical($G$)): $\frac{1}{60}$

### A.4.6 Additional Conjectures

The following conjectures are just a couple of the conjectures that had code that was unable to be run but are still potentially interesting from Claude:

**Conjecture A.13.** *Let $G$ be a non-solvable group. For any element $x$ in $G$, if $\mathrm{Sol}_G(x)$ is a proper subgroup of $G$, then the intersection of $\mathrm{Sol}_G(x)$ with all of its conjugates in $G$ is always contained in the hypercenter of $G$.*

**Conjecture A.14.** *Let $G$ be a non-solvable group. For any element $x$ in $G$, if $\mathrm{Sol}_G(x)$ is a proper subgroup of $G$, then the normalizer of $\mathrm{Sol}_G(x)$ in $G$ contains at least one element from each non-abelian composition factor of $G$.*

**Conjecture A.15.** *Let $G$ be a non-solvable group. For any element $x$ in $G$, if $\mathrm{Sol}_G(x)$ is a proper subgroup of $G$, then the commutator subgroup $[\mathrm{Sol}_G(x), G]$ contains at least one non-identity element from each non-abelian composition factor of $G$.*

**Conjecture A.16.** *Let $G$ be a non-solvable group. For any element $x$ in $G$, if $\mathrm{Sol}_G(x)$ is a proper subgroup of $G$, then the intersection of $\mathrm{Sol}_G(x)$ with its normalizer in $G$ is always metabelian.*

**Conjecture A.17.** *Let $G$ be a non-solvable group. For any element $x$ in $G$, the subgraph of the power graph of $G$ induced by $\mathrm{Sol}_G(x)$ is always connected.*

### A.5 OpenAI o1

Here we briefly summarize data taken from OpenAI o1(23). Due to initial limiting factors and the short timeline, the authors have not finished gathering sufficient data to be included into the main body. We recieved 51 conjectures with the following data:

Table 3: OpenAI o1 Data

| Category | OpenAI o1 |
| --- | --- |
| Unique Conjectures | 51 |
| Total Output | 51 |
| No Counter-Examples | 22 |
| Couldn't Execute Code | 1 |
| Conjecture Failed | 28 |

Perhaps unsurprisingly, the new model was able to execute code properly roughly 98% of the time. Furthermore, the model only once needed to take a debug step. The authors believe that the conjectures are not any significantly more interesting than the other ones given by Gemini and Claude, although this model does give more interesting conjectures than GPT-4 in terms of the willingness to give conjectures about relation of the solubilizer to other subgroups, conjugacy classes, etc. We note that this model was more willing to stray from the system prompt in the GAP code wherein it generated code that checked non-solvable groups that were not simple. Lastly, observe that all of the conjectures are unique, which leaves this model so far being (albeit with a lower than desireable number of samples to make a strong claim) the best one to use for this task in all aspects. The authors intend to explore this further in the coming weeks.

#### A.5.1 Examples:

The following are four randomly chosen examples and their output.

**Conjecture A.18.** *In any finite non-solvable group $G$, the intersection of $Sol_G(x)$ over all $x \in G \setminus R(G)$ is equal to $R(G)$, the solvable radical of $G$.*

**Output A.19.** *No Counter-examples!*

**Conjecture A.20.** *In any finite non-solvable group $G$, for any element $x \in G \setminus R(G)$, there does not exist an element $y \in Sol_G(x)$ such that $\langle x, y \rangle = G$.*

**Output A.21.** *No Counter-examples!*

**Conjecture A.22.** *In any finite non-solvable group $G$, for any elements $x, y \in G \setminus R(G)$, if $y \in Sol_G(x)$, then $x \notin Sol_G(y)$.*

**Output A.23.** *Conjecture failed for group: $A_5$*

**Conjecture A.24.** *In any finite non-solvable group $G$, for any elements $x, y \in G \setminus R(G)$, if $\langle x, y \rangle$ is solvable, then $x$ and $y$ are both contained in a common solvable maximal subgroup of $G$.*

**Output A.25.** *Conjecture failed for group: $PSL(3, 2)$*

The system failed to write code for a single conjecture:

**Conjecture A.26.** *In any finite group $G$, the solubilizer $\mathrm{Sol}_G(x)$ is invariant under all automorphisms of $G$ that fix $x$.*

### A.6 Conjecture 'Interestingness'

In mathematics, evaluating the "interestingness" of a conjecture or problem is inherently subjective and resists quantification. However, if one seeks a quantitative approach, there are several aspects to be considered: the conjecture's depth, its generality or specificity, its implications for other fields, simplicity of solution, and whether it leads to significant advancements or novel methods. Indeed, some of these are impossible to predict, and are not always necessary for a conjecture to be labeled interesting. Some conjectures, like the Riemann Hypothesis, have clear applications to a wide range of fields, yet others garner interest without obvious practical use. Consider the Collatz

Conjecture: despite its straightforward formulation, the conjecture resists resolution and has few known applications, yet it draws wide attention due to its seemingly simple, though elusive nature. Furthermore, a conjecture's "interestingness" often depends on historical context, cultural influence within mathematical communities, and its perceived difficulty or elegance.

Another difficulty in quantifying interestingness is the risk of conflating technical complexity with profundity. A conjecture could be formally intricate yet lack broader appeal or connection to other domains. Additionally, highly specialized conjectures may be overlooked by non-specialists despite their beauty or importance for those knowledgebale in the field. Further, the evolving nature of mathematical interest itself becomes an issue; conjectures once regarded as obscure can gain recognition as foundational connections become clearer.

Nevertheless, certain conjectures seem almost universally intriguing. The Poincaré Conjecture and Fermat's Last Theorem captivated broad attention due to their simplicity, profound implications, and historical legacy. If just pieces of these ideas could somehow be built into a standardized metric, many studies will surely benefit.

## A.7 Similarity Analysis

We include figures that show the similarity heatmaps between conjectures below. It is visually apparent from these maps that Claude in general had the most syntactic similarity. Indeed, with conjectures exampled as the following A.27A.28, the entire structure of the first conjecture is held within the second but they are not the same idea.

**Conjecture A.27.** *"Let $G$ be a finite non-solvable group. Then for any element $x$ in $G$, if $\mathrm{Sol}_G(x)$ is a proper subgroup of $G$, there exists a prime $p$ dividing $|G|$ such that $\mathrm{Sol}_G(x)$ intersects at least two distinct Sylow $p$-subgroups of $G$ non-trivially."*

**Conjecture A.28.** *Let $G$ be a finite non-solvable group. Then for any element $x$ in $G$, if $\mathrm{Sol}_G(x)$ is a proper subgroup of $G$, there exists a prime $p$ dividing $|G|$ such that $\mathrm{Sol}_G(x)$ intersects at least two distinct Sylow $p$-subgroups of $G$ non-trivially, **but does not contain any full Sylow $p$-subgroup of $G$**.*

GPT-4 clearly has the most syntactic differences in the conjectures. While many of the conjectures reference the same idea, the way that they are stated is highly variable. Claude and Gemini both have a more methodical approach which show up as lighter colored squares. One can see that Gemini also had a period of runtime where the conjectures were mostly structured similarly with different modifiers at the end of the conjecture. The authors are unsure why these structural 'loops' seem to occur periodically throughout the repeated process, but they are interesting to note regardless. In these patches there didn't seem to be a significant difference in the quality of conjecture.

With regards to the similary with the literature, non-surprisingly GPT-4 had the highest similarity due to the reproduction of a conjecture from literature as noted above. Otherwise, the distinction in the models between themselves was roughly comparable with that in literature. We note that Claude had the lowest maximum similarity, that all of the models had a minimum similarity of zero, and that on average, the models were slightly more disjoint from literature than they were from each other.

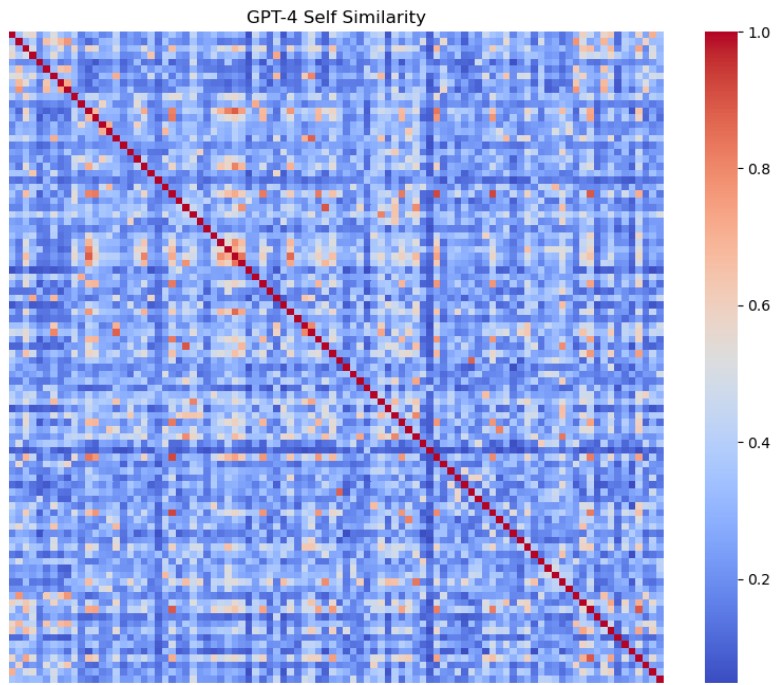

Figure 2: GPT-4 Cosine Self Similarity

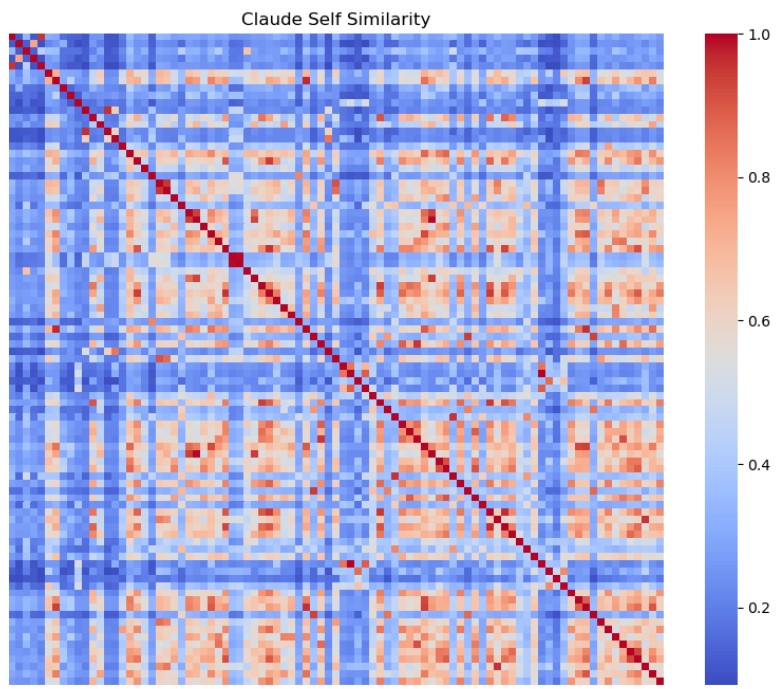

Figure 3: Claude Cosine Self Similarity

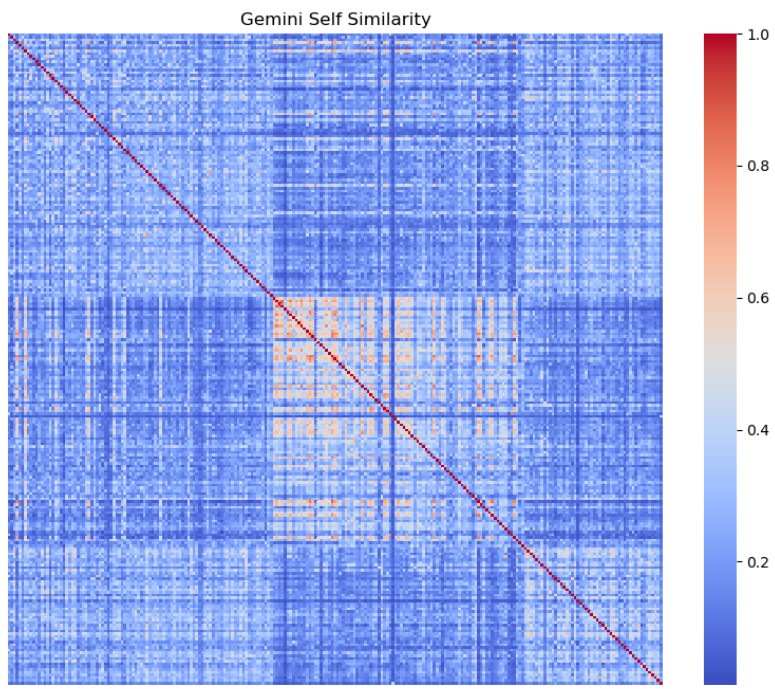

Figure 4: Gemini Cosine Self Similarity

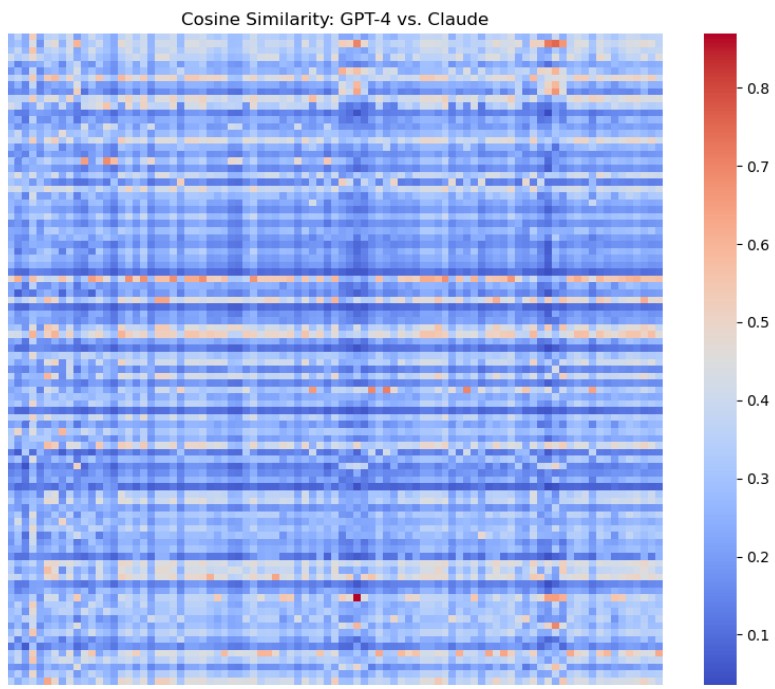

Figure 5: GPT vs. Claude Cosine Similarity

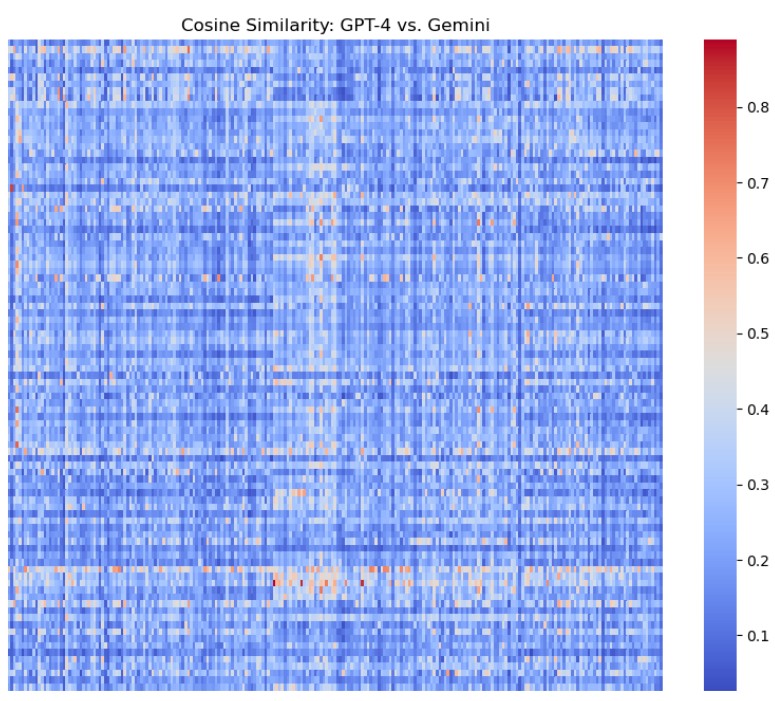

Figure 6: GPT vs. Gemini Cosine Similarity

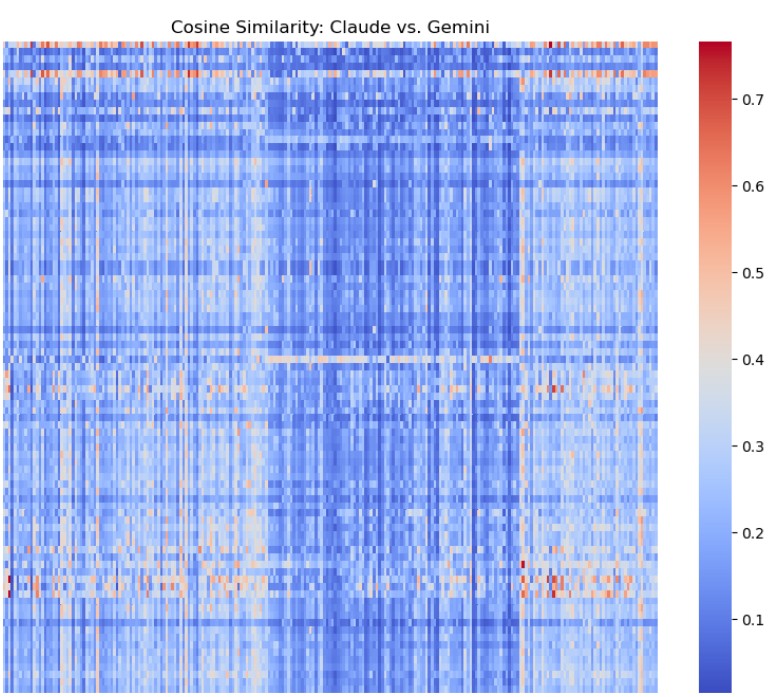

Figure 7: Claude vs. Gemini Cosine Similarity

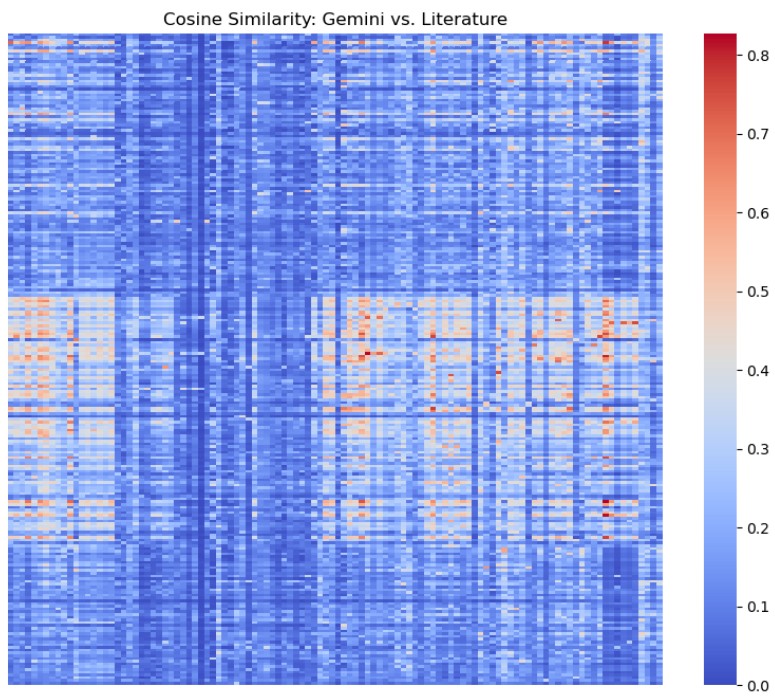

Figure 8: Gemini vs. Literature Cosine Similarity

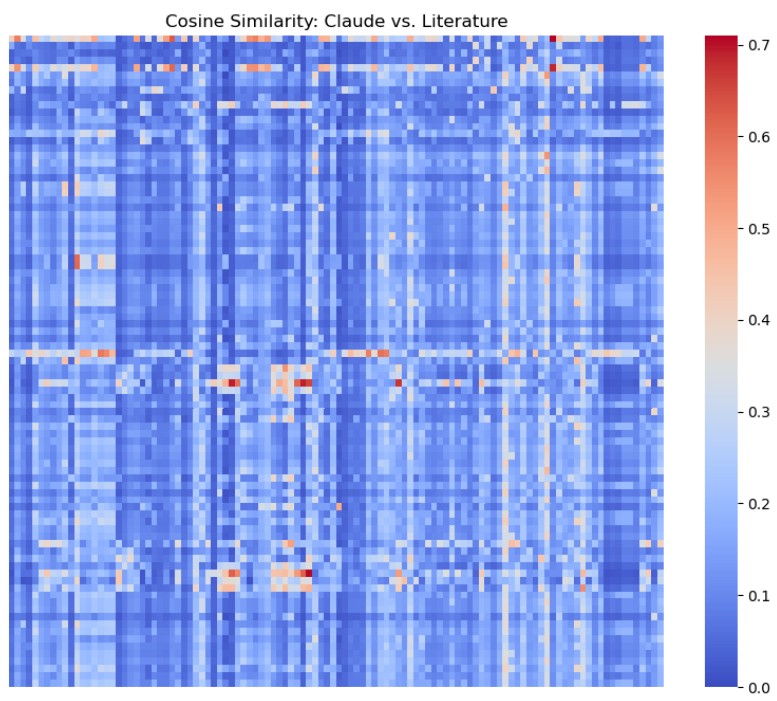

Figure 9: Claude vs. Literature Cosine Similarity

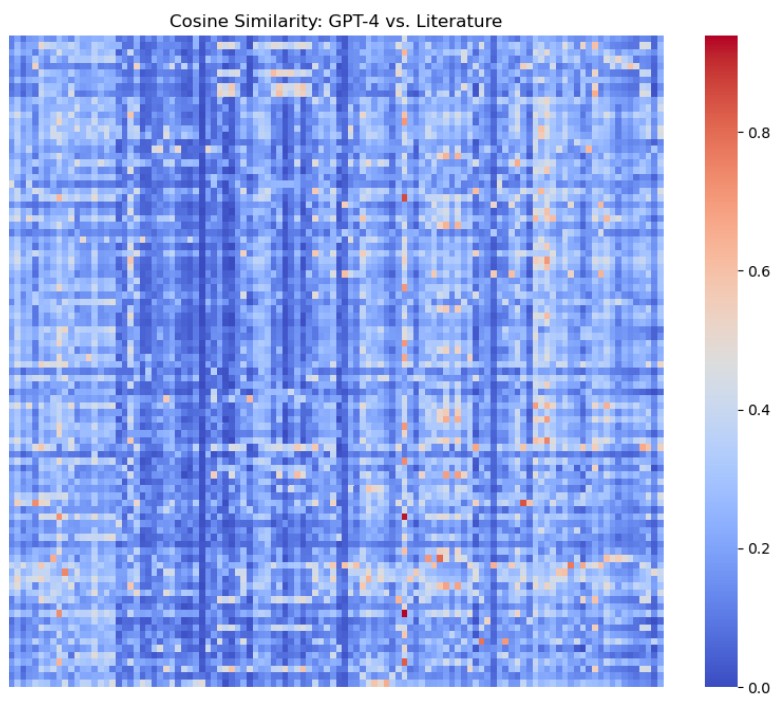

Figure 10: GPT-4 vs. Literature Cosine Similarity

