# OpenReview forum: "Mining Math Conjectures from LLMs: A Pruning Approach"
_NeurIPS.cc/2024/Workshop/MATH-AI — MATH-AI 24_

### Official Review · Reviewer_Hfos · 2024-09-29
**Evaluating LLMs for Mathematical Conjecture Generation**

**Rating:** 8
**Confidence:** 4

**Review:**

This paper presents an interesting approach to leveraging large language models (LLMs) for generating and evaluating mathematical conjectures, focusing on the domain of group theory and the concept of solubilizers. The authors use ChatGPT, Gemini, and Claude to generate conjectures about solubilizers and then employ the GAP computer algebra system to test these conjectures on non-solvable groups.

Pros:
1. Novel application: The paper explores an innovative use of LLMs in mathematics, specifically for conjecture generation, which is an under-explored area.
2. Methodology: The authors present a clear methodology for generating and pruning conjectures using LLMs and computational tools.
3. Comparative analysis: The study compares the performance of multiple LLMs (ChatGPT, Gemini, Claude), providing insights into their relative strengths and weaknesses.
4. Domain selection: Focusing on solubilizers, a relatively new concept in group theory, helps ensure the originality of the generated conjectures.
5. Practical implications: The approach could potentially assist mathematicians in exploring new areas and generating novel hypotheses.

Cons:
1. Limited scope: The study focuses solely on group theory and solubilizers, which may limit the generalizability of the findings to other areas of mathematics.
2. Sample size: The number of generated conjectures (430 unique outputs) is relatively small, which may affect the statistical significance of the comparisons between models.
3. Evaluation criteria: The paper lacks a clear framework for evaluating the quality and novelty of the generated conjectures beyond their immediate falsifiability.
4. Human expert validation: There is no mention of human mathematicians evaluating the generated conjectures for their potential significance or novelty.
5. Limited discussion of practical applications and broader impact: The authors do not sufficiently explore the potential real-world applications and broader implications of their work. While they mention that this approach could assist mathematicians, they do not delve into how it might transform mathematical research practices, accelerate discoveries in other fields that rely on mathematical foundations, or impact mathematics education. The paper would benefit from a more thorough discussion of how this AI-driven approach to conjecture generation could influence the broader scientific community, potentially leading to breakthroughs in related fields such as physics, computer science, or cryptography. Additionally, the authors could have addressed how this technology might be integrated into mathematical software tools or research workflows, and its potential to democratize mathematical discovery by making it more accessible to researchers with varying levels of expertise.

General Comments:
- The paper demonstrates that LLMs can indeed generate plausible mathematical conjectures and, in some cases, produce original ideas that warrant further investigation. The methodology of using LLMs for generation and computational tools for validation is sound and could be extended to other areas of mathematics.
- The authors provide a thorough analysis of the performance of different LLMs, highlighting their strengths and limitations. For instance, ChatGPT showed better performance in generating conjectures that could not be immediately falsified, while Claude was more effective at identifying counterexamples.
- However, the paper would benefit from a more rigorous evaluation of the quality and potential impact of the generated conjectures. Involving human experts to assess the most promising conjectures could provide valuable insights into the practical utility of this approach.
- The discussion of the limitations of LLMs in executing code and the high percentage of non-executable code (63.02% of unique outputs) is particularly insightful. This highlights an area for improvement in LLM training and prompts further research into enhancing their code generation capabilities.
- The authors' suggestion for future work, including the integration of automated theorem provers or neuro-symbolic proof engines, is promising and could lead to more robust end-to-end systems for conjecture generation and proof.
- In conclusion, while the paper has some limitations, it presents a novel and potentially impactful approach to using AI in mathematical research. The methodology and findings contribute to our understanding of how LLMs can be leveraged in creative mathematical tasks and open up new avenues for AI-assisted mathematical discovery.

Overall, this paper presents a novel and interesting approach to using LLMs for mathematical conjecture generation. While it has some limitations, the methodology is sound and the results are promising, warranting acceptance and further exploration in this direction.

---

### Official Review · Reviewer_kMbV · 2024-10-07
**Interesting approach to generate conjectures in algebra**

**Rating:** 6
**Confidence:** 4

**Review:**

Summary:
The paper introduces a novel approach for generating mathematical conjectures by combining LLMs with the GAP computer algebra system. The paper explores how LLMs can suggest conjectures, and how GAP is used to find counter-examples to these conjectures.

Strengths:
- An interesting approach, combining LLMs with GAP code.
- Clear writing style.

Opportunities for improvements:
- I was hoping the authors could discuss both the diversity of the generated conjectures, and the limitations of the approach in a more quantitative way: How diverse are the final conjectures? Does scaling the approach allow generating more and more conjectures or does the process reach a point of diminishing returns?
- The scope of this paper appears to be limited to conjectures in the field of algebra, as the approach uses GAP, which can only be used in algebra. A discussion on a possible generalization of the approach to other fields could be interesting.

Final remark:
The approach introduced in this paper is interesting, and I believe the workshop can benefit from the paper discussion. However, given the limitations of the approach (diversity, scaling, limited to algebra), I opt to give a rating of 6.

---

### Official Review · Reviewer_6eaG · 2024-10-08
**Rigorous Experimental Design**

**Rating:** 6
**Confidence:** 3

**Review:**

This study explores the integration of large language models (LLMs) with computational tools like GAP to successfully generate and test original mathematical conjectures. The results demonstrate the potential of combining these technologies to make significant strides in mathematical discovery. The models show promise, although there are minor challenges in generating executable code and avoiding duplicate conjectures.

The strengths of this paper lie in its rigorous experimental design and the substantial amount of testing conducted, which thoroughly validates the approach. However, one notable weakness is the lack of clarity on the significance of creating original mathematical conjectures and why this area is worth exploring. The paper’s contribution to the field also feels somewhat ambiguous and would benefit from a more detailed explanation of its impact and relevance

---

### Decision · Program_Chairs · 2024-10-09

**Decision:**

Accept

**Comment:**

Reviewers find the approach of generating conjectures using LLMs and GAP interesting and novel. The paper could improve by having more comprehensive analysis and discussion of the results, but seems to be a worthwhile contribution to the workshop.